ᵃ | **Open Peer Review** | Environmental Microbiology | Research Article

# Diverse winter communities and biogeochemical cycling potential in the under-ice microbial plankton of a subarctic river-to-sea continuum

Marie-Amélie Blais,[1,2,3,4] Warwick F. Vincent,[1,2,3,4] Adrien Vigneron,[1,2,3,4] Aurélie Labarre,[1,2,4,5] Alex Matveev,[1,3,4] Lígia Fonseca Coelho,[6,7,8] Connie Lovejoy[1,2,4,5]

**ABSTRACT**   Winter conditions greatly alter the limnological properties of lotic ecosystems and the availability of nutrients, carbon, and energy resources for microbial processes. However, the composition and metabolic capabilities of winter microbial communities are still largely uncharacterized. Here, we sampled the winter under-ice microbiome of the Great Whale River (Nunavik, Canada) and its discharge plume into Hudson Bay. We used a combination of 16S and 18S rRNA gene amplicon analysis and metagenomic sequencing to evaluate the size-fractionated composition and functional potential of the microbial plankton. These under-ice communities were diverse in taxonomic composition and metabolically versatile in terms of energy and carbon acquisition, including the capacity to carry out phototrophic processes and degrade aromatic organic matter. Limnological properties, community composition, and metabolic potential differed between shallow and deeper sites in the river, and between fresh and brackish water in the vertical profile of the plume. Community composition also varied by size fraction, with a greater richness of prokaryotes in the larger size fraction (>3 µm) and of microbial eukaryotes in the smaller size fraction (0.22–3 µm). The freshwater communities included cosmopolitan bacterial genera that were previously detected in the summer, indicating their persistence over time in a wide range of physico-chemical conditions. These observations imply that the microbial communities of subarctic rivers and their associated discharge plumes retain a broad taxonomic and functional diversity throughout the year and that microbial processing of complex terrestrial materials persists beneath the ice during the long winter season.

**IMPORTANCE**   Microbiomes vary over multiple timescales, with short- and long-term changes in the physico-chemical environment. However, there is a scarcity of data and understanding about the structure and functioning of aquatic ecosystems during winter relative to summer. This is especially the case for seasonally ice-covered rivers, limiting our understanding of these ecosystems that are common throughout the boreal, subpolar, and polar regions. Here, we examined the winter under-ice microbiome of a Canadian subarctic river and its entry to the sea to characterize the taxonomic and functional features of the microbial community. We found substantial diversity in both composition and functional capabilities, including the capacity to degrade complex terrestrial compounds, despite the constraints imposed by a prolonged seasonal ice-cover and near-freezing water temperatures. This study indicates the ecological complexity and importance of winter microbiomes in ice-covered rivers and the coastal marine environment that they discharge into.

**KEYWORDS**   winter limnology, coastal water, river, metagenome, microbiome, prokaryotes, microbial eukaryotes, subarctic, under-ice, size fraction

Address correspondence to Marie-Amélie Blais, marie-amelie.blais@usherbrooke.ca.

The authors declare no conflict of interest.

See the funding table on p. 18.

Winter conditions and seasonal ice cover profoundly alter the limnological properties of aquatic ecosystems, but the consequences for their microbial communities are poorly understood, especially for ice-covered rivers and streams. In winter, the water flow in non-regulated rivers is typically lower than at other times of year, reducing sediment transport and erosion (1). Temperatures are also lower, reducing microbial metabolism and growth rates (2). In addition, inputs from the frozen watershed, exchanges with the atmosphere, and light availability in the water column are all reduced, acting to diminish substrate and energy supply to the microbial communities (3).

Winter under-ice pelagic microbial communities can differ markedly from those in summer open-water conditions (4, 5), with some microbial taxa that excel during winter, while others decrease their activity or enter into dormancy. Seasonal changes in bacterial communities can vary according to lifestyle, with evidence that free-living planktonic bacteria may have greater temporal stability and lower dissimilarity between seasons than particle-associated bacteria (6, 7).

Previous studies based on the identification of microbial community composition using amplicon sequencing have inferred that distinct metabolic pathways and energy acquisition mechanisms may be favored by winter conditions or maintained despite these conditions. For example, in the Saint-Charles River (southern Quebec, Canada), the dominant microbial eukaryotes had the potential for phagotrophy and phototrophy (8), while the bacterial community composition suggested active carbon, nitrogen, and iron cycles in winter (7). These biogeochemical cycles, along with sulfur cycling, were also presumed to be active under the ice of coastal lagoons along the eastern Alaskan Beaufort Sea, with low organic carbon inputs during winter and depletion of labile organic matter favoring chemolithoautotrophic taxa (9). In the same study, the microbial eukaryotic community was dominated by heterotrophic and parasitic taxa in winter, with little contribution from phototrophs (9). Metagenomic analysis in the same region further confirmed these inferences, with a greater abundance of genes involved in nitrification, methane metabolism, and chemoautotrophic carbon fixation in winter (10). However, the metabolic potential of winter microbiomes remains largely under-characterized, with few metagenomic studies to date, notably from Lake Baikal (11), temperate and boreal lakes (12), and permafrost thaw lakes (13).

In the present study, we examined the winter under-ice microbiome of a Canadian subarctic river and its plume on entry to the sea. The aim was to characterize both taxonomic and functional attributes of the winter microbiome. We sampled the Great Whale River and its associated plume in late winter when the river and coastal Hudson Bay are covered by their seasonally thickest ice. Size fractionation was used to differentiate the community composition between free-living prokaryotes and picoeukaryotes in a small size fraction (0.22–3 µm) from larger microbial eukaryotes and colonial, filamentous, and particle-associated prokaryotes in a large size fraction (>3 µm). We used amplicon sequencing to determine the community structure and used metagenomic sequencing to assess the metabolic and biogeochemical potential of the community. For the latter, we focused on identifying genes associated with nitrogen and sulfur cycles, carbon metabolism (carbon fixation and methane cycling), and phototrophic activity (photosynthesis and pigments). Additionally, the metagenomes were analyzed to assess the community capacity to degrade terrestrially derived aromatic carbon compounds. We focused on a set of marker genes for pathways involved in the breakdown of these compounds, specifically for pathways that were previously identified in the Canada Basin of the Arctic Ocean (14). We hypothesized that the proportional representation of genes associated with aromatic carbon degradation would be greater in the river, which is directly supplied by terrestrial inputs from the boreal forest-tundra, than in the offshore Arctic marine waters that is far from river source waters.

## MATERIALS AND METHODS

### Study site and sampling

The Great Whale River is a 726-km-long subarctic river flowing across the Canadian Precambrian Shield and discharging into southeastern Hudson Bay, near Whapmagoostui and Kuujjuarapik villages, where it creates an extensive buoyant plume (15). The river is ice covered more than 6 months of the year [from approximately late November to mid-May; (16)]. During this period, the river discharge is reduced, with a minimum in April [200 vs 910 $m^3$ $s^{-1}$ at peak flow in June; (17, 18)]. The Great Whale River watershed has been widely studied and is viewed as a model system for isolated/sporadic permafrost and a sentinel for global change (18). More recently, the microbial communities of the river and its plume during the summer season (19, 20) have been examined, and these results showed strong gradients in community structure across the freshwater-saltwater transition. A study on the winter coastal ice and underlying seawater (21) in this area indicated diverse bacterial communities but did not extend upstream or include metagenomic analysis.

To encompass a broad range of conditions, 12 samples were collected along a 10-km downstream transect of the Great Whale River and its plume into coastal Hudson Bay. Sampling was in late winter, from 25 February to 1 March 2019. At the time of the sampling, ice thickness ranged from 73 to 110 cm in the river and was 71 cm in the plume region (Table 1). Snow cover was variable on the coastal bay ice, and there was no snow on the ice where the plume was sampled. Snow depths ranged from 11 to 53 cm at the river sites. At each site, holes were bored through the ice, and water was collected directly beneath the ice using a Kemmerer water sampler. The collected water was then transferred to 20 L acid-washed and sample-rinsed LDPE Cubitainers. The vertical salinity structure of the plume was determined before sampling, and more saline (termed brackish) water samples were collected from 4 m (Fig. 1a). Samples from similar habitats were grouped: the three upstream sampling points in the river that were characterized by a shallow under-ice water depth (<1 m) and were collected 5–10 m apart (*RSh* sites); the three river samples for which the water depth was greater were collected ~2 km apart (*R* sites); the three plume samples collected under the ice (*PS* sites) and the three brackish samples at 4 m depth (*P4M* sites; Fig. 1b; both groups were sampled from separate bottle casts in the same hole in the ice).

Physico-chemical profiling at each of the sampling sites was obtained with an RBR Concerto conductivity, temperature, depth (CTD) logger, and a YSI EXO2 multiparameter probe (salinity, temperature, and dissolved oxygen). The profiles were not measured for sites *R.2* and *R.3* due to logistical constraints. Water samples for chemical analyses were filtered and subsampled at the Centre for Northern Studies (CEN) research

**TABLE 1** Selected environmental properties along the Great Whale River and plume (mean with SD in parentheses; $n = 3$ unless stated otherwise)[a]

|  | *RSh* | *R* | *PS* | *P4M* |
|---|---|---|---|---|
| Temperature (°C) | 0[b] | 0[b] | 0[b] | −0.95[b] |
| Snow depth (cm) | 46.3 (5.8) | 22.7 (13.9) | 0[b] | 0[b] |
| Ice thickness (cm) | 95 (8.7) | 87 (20.1) | 71[b] | 71[b] |
| $O_2$ (mg $L^{-1}$) | 15.65[b] | 16.34[b] | 16.73[b] | 15.45[b] |
| Specific conductivity (µS $cm^{-1}$) | 51[b] | 38[b] | 682[b] | 36,010[b] |
| DIC (mM) | 0.11 (0.02) | 0.07 (0.01) | 0.09 (0.02) | 1.56 (0.02) |
| DOC (mg C $L^{-1}$) | 6.2 (1.0)[c] | 3.1 (0.1) | 3.5 (0.1) | 2.7 (0.3) |
| $a_{320}$ ($m^{-1}$) | 19.6[b] | 12.7 (0.01)[c] | 12.7 (0.1) | 6.3 (1.1)[c] |
| $SUVA_{254}$ (L mg $C^{-1}$ $m^{-1}$) | 3.93[b] | 4.54 (0.12)[c] | 3.99 (0.14) | 2.80 (0.06)[c] |
| $S_R$ | 0.89[b] | 0.88 (0) | 0.89 (0)[c] | 1.01 (0.01)[c] |
| $S_{289}$ | 0.02[b] | 0.02 (0)[c] | 0.02 (0) | 0.02 (0)[c] |

[a]*RSh* corresponds to shallow river sites (depth <1 m); *R*, deeper river sites (depth >1 m); *PS*, plume surface; *P4M*, plume at 4 m depth, DIC, dissolved inorganic carbon; DOC, dissolved organic carbon.
[b]$n = 1$.
[c]$n = 2$.

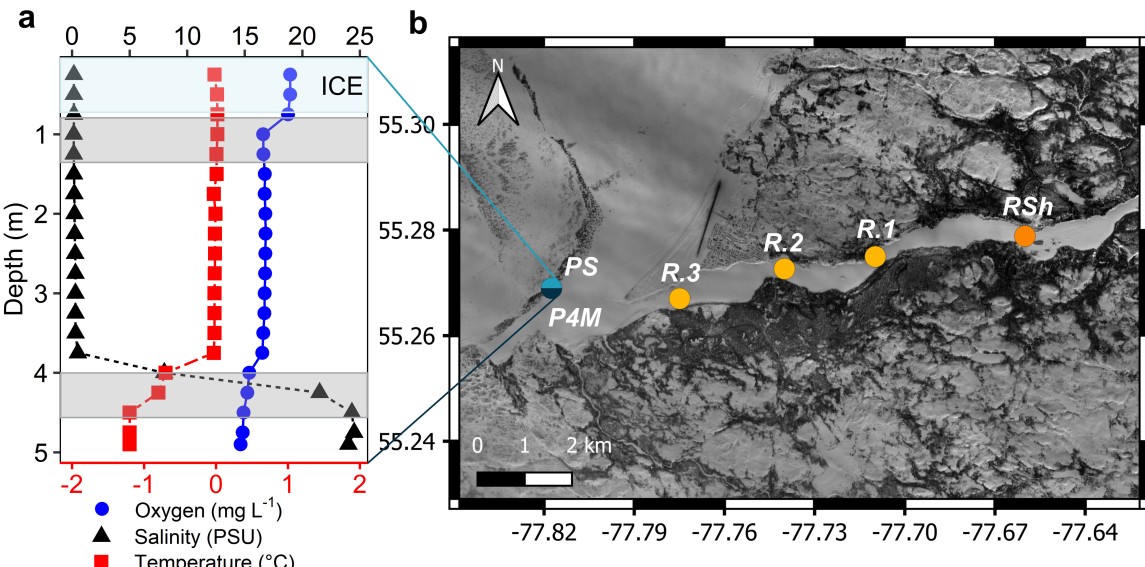

**FIG 1** (a) Vertical profile of temperature (bottom *x*-axis), oxygen, and salinity (top *x*-axis for both) in the plume. The upper blue-shaded band corresponds to the ice cover and two gray bands to the depth strata that were sampled for microbial analysis. (b) Map of the sampling sites along the Great Whale River and its plume in Hudson Bay (Copernicus Sentinel-2 data 2019, processed by ESA).

station in Whapmagoostui-Kuujjuarapik within hours of sampling and then shipped to Laval University and INRS (Quebec City, Canada) for final analysis. Water was filtered until clogging onto pre-combusted, pre-weighed 47 mm GF/F filters (0.7 µm) for total suspended sediments (TSS) and onto 0.7 µm GF/F filters for chlorophyll-*a* concentrations. Water samples for dissolved organic carbon (DOC) and colored dissolved organic matter (CDOM) concentrations were filtered through pre-rinsed 0.2 µm cellulose acetate filters (Advantech MFS).

## Laboratory manipulations and analysis

Total phosphorus (TP, detection limit 2 µg P L$^{-1}$) and total nitrogen (TN, detection limit 10 µg N L$^{-1}$) concentrations were determined on unfiltered acidified (H$_2$SO$_4$, 0.1% final concentration) water samples using, respectively, the ascorbic acid colorimetric method (Standard Methods 4500-PE) and the hydrazine reduction-sulfanilamide colorimetric method (Standard Methods 12-107-04-1-E). Dissolved inorganic carbon (detection limit 0.02 mM) concentration was determined on unfiltered acidified (HCl, 0.05 M) water samples using the headspace gas chromatography method. TSS concentration was determined by weighing dried filters (80°C for 44 hours). Chlorophyll-*a* was extracted with 95% MeOH, and the concentrations were determined by High-Pressure Liquid Chromatography (HPLC) as previously described (22). Technical duplicates of unfiltered water samples preserved in glutaraldehyde (1% final concentration) were used to determine bacteria and phytoplankton cell concentrations with a BD Accuri C6 flow cytometer (BD Biosciences) as previously described (20).

DOC samples were acidified and analyzed by high-temperature catalytic oxidation with non-dispersive infrared detection method using a Shimadzu VCPH analyzer (Standard Methods 5310 B). Two blanks consisting of filtered ultrapure water were analyzed along with the samples to correct for any DOC release from the filters. The mean value of the blanks was subtracted from the results. CDOM absorbance was measured in a LAMBDA 850 UV/Vis Spectrophotometer (Perkin Elmer). Before conversion to absorption coefficients, spectra were blank corrected using ultrapure water, and null-point adjustments were made using the mean value from 750 to 800 nm.

The absorption coefficient at 320 nm ($a_{320}$, m$^{-1}$) was used as CDOM concentration proxy, and the specific ultraviolet absorbance at 254 nm per unit DOC (SUVA$_{254}$, L mg

$C^{-1}$ m$^{-1}$) was used as a proxy for organic matter aromaticity (23). The abs_parms function of the staRdom package (24) for R was used to calculate CDOM spectral slopes [$S$; (25)]. Spectral slope for the wavelength interval 279–299 nm ($S_{289}$) was used as an index of autochthonous carbon (25, 26) and the spectral slope ratio ($S_R$, $S_{275-285}/S_{350-400}$) as an indicator of molecular weight (27). Some bottles were damaged during transport, and DOC data are therefore lacking for site *RSh.1* and CDOM data for sites *RSh.1*, *RSh.3*, *R.2*, and *P4M.1*. The HPLC, physical, and chemical data are available in the northern environmental data repository *Nordicana D* (28).

## Microbial sample processing and analyses

In the field laboratory, the samples for amplicon analysis were filtered sequentially through a 3-µm pore size, 47-mm-diameter polycarbonate filter (large fraction), and 0.22-µm Sterivex filter unit (Millipore; small fraction) using a peristaltic pump. Samples for metagenomic analysis were directly filtered through a 0.22-µm Sterivex filter unit (Millipore). Filters were preserved in RNA*later* solution (Invitrogen) and stored at −60℃ for a week at the CEN research station and then at −80℃ until DNA extraction. All-Prep DNA/RNA Mini Kit (Qiagen) was used to extract DNA as previously described (20). Sample-free controls were processed as regular samples to remove the potential contaminants from the final amplicon sequence variant (ASV) table.

Microbial community composition was determined on both size fractions by amplification and sequencing of the V4 region of the 16S rRNA and 18S rRNA genes. Primers E572F/E1009R (29) were used for microbial eukaryotes and 515F(Parada)/806R (Apprill) (30, 31) for prokaryotes (Bacteria and Archaea). Two-step PCR was performed to amplify the gene and to add a sample index and Illumina MiSeq adapters. Both PCR reaction mixes and PCR conditions were as previously described (20), except the first PCR for prokaryote samples from the 3-µm filter for sites *R.1*, *PS.1*, and *P4M.1*, for which 3 µL of DNA template was used, instead of 1 µL, to ensure amplification. After each step, PCR products were purified using magnetic beads (sparQ PureMag Beads, Quantabio) after verification on a 1% agarose gel. Final PCR products were pooled equimolarly and purified, separately for 16S and 18S rRNA genes, after quantification on a Qubit 2.0 Fluorometer (Life Technologies) and quality control on a Spark multimode microplate reader (Tecan). Sequencing was performed at the Plateforme d'analyses génomiques (IBIS, Laval University) on an Illumina MiSeq system (2 × 300 bp). The 3-µm filter sample of site *R.3* was removed as the amplification was unsuccessful for both the prokaryote and microbial eukaryote primers.

To produce an ASV table, reads were imported into QIIME2 [v.2020.8; (32)] after a quality check using FastQC [v.0.11.8; (33)]. The denoising and merging of the reads, and chimeras removing were made using DADA2 (34). Taxonomic assignments were made using the Bayesian classifier implemented on Mothur [v.1.41.3; (35)] with the Silva database [v.138.1; (36, 37)] for bacteria and the PR$^2$ database [v.4.12.0; (38)]) for microbial eukaryotes. For the final 18S ASV table, we removed ASVs identified as unknown, Embryophyceae, metazoa, and Rhodophyta. Unclassified eukaryote taxa were verified with the Silva database, and ASVs identified as bacteria were also removed. For the final 16S ASV table, we removed ASVs identified as unknown, eukaryotes, mitochondria, and chloroplasts. Singletons were removed from both ASV tables. After the quality filtering, we retained 1,002,943 reads (43,606 ± 13,208, mean ± SD, reads per sample; $n$ = 23) for prokaryotes (Bacteria and Archaea) and 426,447 reads (18,541 ± 5,758; $n$ = 23) for microbial eukaryotes.

Metagenomic libraries were prepared on the 12 0.22-µm filter samples using a Nextera XT Library Kit (Illumina). These were pooled equimolarly and sequenced in two Illumina MiSeq (2 × 300 bp) runs at the Plateforme d'Analyses Génomiques (IBIS, Laval University). Data sets from the two runs were pooled by sample and quality filtered using Trimmomatic [v.0.39; (39)] after quality check using FastQC [v.0.11.8; (33)]. After quality filtering, we retained 90,341,424 reads (7,528,452 ± 1,339,597 reads per sample, $n$ = 12). Analysis was then done using the SqueezeMeta pipeline [v1.5.1; (40)]. A co-assembly

of the 12 metagenomes was done using SPAdes (41) with kmer size of 21, 33, 55, 77, 99 pb. Prinseq (42) was used to remove short contigs (<200 bps) and to obtain contig statistics. Open reading frames (ORFs) were predicted using Prodigal (43) with additional ORFs obtained by Diamond BlastX (44). Homology searching against GenBank (45) for taxonomic annotation and the KEGG database (46) for the functional annotation were done using Diamond (44). Read mapping against contigs was performed using Bowtie2 (47) and was used to estimate the abundance of genes in each metagenome. To account for possible bias in the coassembly, functional assignments of the reads were also performed using Diamond (44) against GenBank (45) and KEGG (46). Metabolic pathways, modules, and reactions in which our KOs (KEGG orthology) were assigned were determined using KEGG mapper [v.5.0; (48, 49)]. To be considered present, KOs had to be present in both data sets. KOs of biogeochemical cycles and metabolic pathways of interest (nitrogen, sulfur, and carbon metabolism; photosynthesis and pigments; and aromatic compound degradation) that were detected only through direct annotation of the reads are mentioned. Unless otherwise noted, all figures representing the metabolic potential of the community were made using the coassembly values. Raw sequences are available at the NCBI Sequence Read Archive (https://www.ncbi.nlm.nih.gov/bioproject) under the bioproject PRJNA999265 for the amplicon and PRJNA999354 for the metagenomes.

## Statistical analyses

Statistical analyses were performed using R [v.4.2.0; (50)] in the RStudio environment [v.2002.2.2.485; (51)] and calculated separately for the microbial eukaryotes and the prokaryotes for the amplicon data sets. A total-sum scaling transformation was applied on the ASV tables prior to statistical analysis, except for alpha diversity, and on the KOs table obtained through direct annotation of the reads. KOs obtained through coassembly were normalized by dividing the number of reads for each KO by the number of reads assigned to the *recA* single-copy gene (K03553). Alpha diversity (Chao1 index) was calculated with the estimate_richness function of the phyloseq package [v.1.40.0; (52)] on ASV tables prior to the removal of singletons. To determine if Chao1 index differed by size fractions, Wilcoxon signed-rank test for paired samples was calculated with wilcox.test function (stats package) after the removal of sample *R.3*, as the large size fraction of this sample did not amplify. Ward hierarchical clustering (ward.d2 option in hclust function, stats package) based on Bray-Curtis dissimilarity [vegdist function, vegan package, v.2.6–4; (53)] was calculated at ASV level to visualize the taxonomic beta diversity among sampling groups and size fractions and, at KOs level, to visualize the functional beta diversity among sampling groups (from the reads and the coassembly). To evaluate if the microbial community composition differed by size fraction and by sampling group and if the community functional potential differed by sampling group (from the reads and the coassembly), a permutational multivariate analysis of variance was calculated (permutational MANOVA, 9,999 permutations, adonis2 function, vegan package) after the verification of the homogeneity of group dispersions (between size fractions and between sampling group, median based, function betadisper and permutest, 9,999 permutations, vegan package). To identify KOs that were differentially abundant along the river between the shallow and the deeper sites (*RSh* vs *R*), and in the vertical plume profile between surface water and brackish water at 4 m depth (*PS* vs *P4M*), a differential abundance analysis was calculated using the DESeq2 package [v.1.36.0; (54)] on the raw abundance of KOs from the coassembly (Wald test). KOs were considered differentially abundant if the adjusted *P*-values were ≤0.01 (Benjamini-Hochberg correction). The *z*-score (standard deviation from the row mean, calculated from normalized gene abundance reads/*recA* reads) was calculated for KOs differentially abundant ($P_a \leq 0.01$) that were involved in pathways, reactions, and modules discussed in the Results and visualized using ggplot2 [v.3.3.6; (55)]. Constrained ordinations (e.g., redundancy analysis) were not performed due to missing limnological variables for some

samples and high correlations among the remaining variables suggesting that they are confounding (Fig. S1).

## RESULTS

### Environmental characteristics

Along the river, the limnological variables (Table 1; Fig. 2) differed between samples collected at sites with shallow water under the ice (sites *RSh.1* = 1 m depth, *RSh.2 and RSh.3* estimated at ≤1 m; hereafter referred to as shallow river samples/sites) and those with deeper water (*R.1* = 8.9 m, *R.2* estimated at >1 m, and *R.3* = 5.3 m; hereafter referred to as deeper river samples/sites). Shallow sites had higher DOC, CDOM ($a_{320}$), TP, TSS, phytoplanktonic cell and chlorophyll-*a* concentrations, and slightly higher specific conductivity.

The vertical profile (Fig. 1a) in the plume revealed a transition from cold, fresh, river water to brackish, slightly colder, Hudson Bay water at 4 m depth. Limnological variables in the freshwater plume (hereafter referred to as plume samples/sites) were similar to those at the deeper water river sites, except for specific conductivity, TP, and bacterial cell concentrations, which were higher in the plume. The brackish Hudson Bay (hereafter referred to as brackish samples/sites) water had lower cell and CDOM concentrations. This CDOM had a lower molecular weight (as indicated by $S_R$ ratio) and aromaticity ($SUVA_{254}$). However, TP, TSS, and chlorophyll-*a* concentrations were higher than those in the freshwater plume.

### Community composition and diversity

Hierarchical clustering revealed that the community composition of prokaryotes (Fig. 3a) and microbial eukaryotes (Fig. 4a) at the ASV level differed by size fraction and

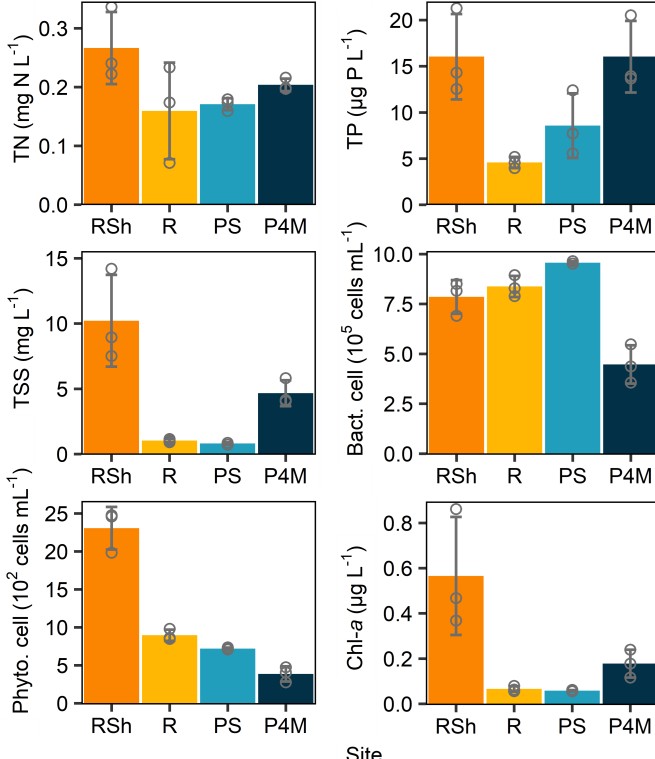

**FIG 2** Means of selected limnological variables for each group of sites. *RSh* corresponds to shallow river sites (depth <1 m); *R*, deeper river sites (depth >1 m); *PS*, plume surface; *P4M*, plume at 4 m depth (brackish water). Circles correspond to individual values, and error bars are SD. *n* = 3.

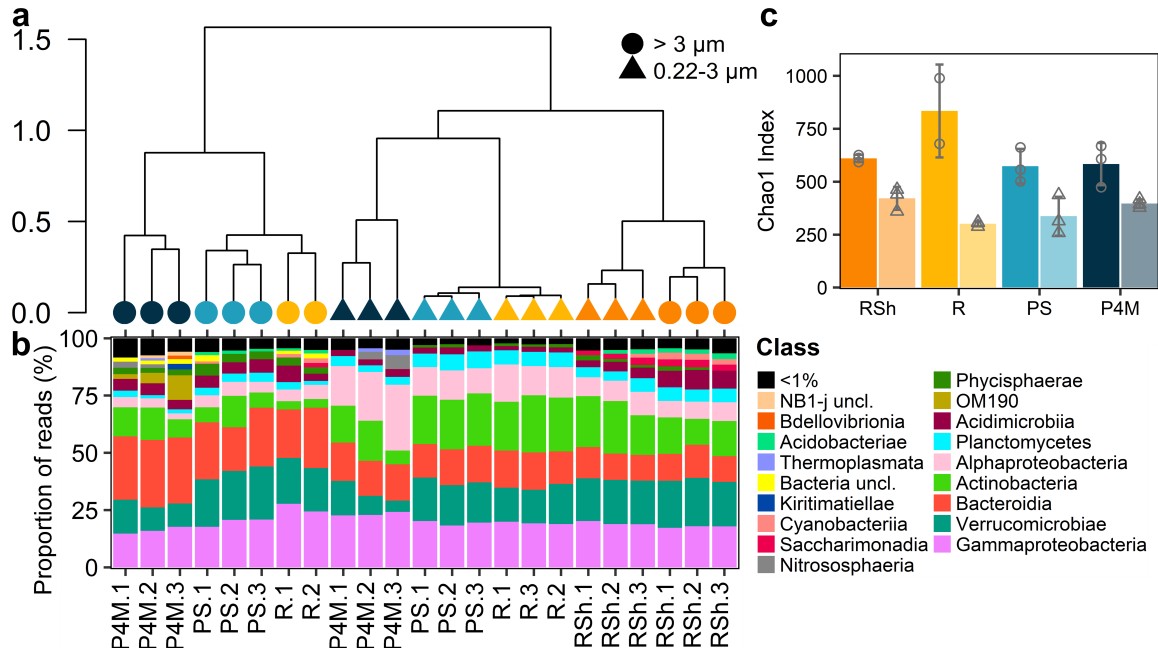

**FIG 3** Prokaryotic communities in the Great Whale River and Plume. (a) Ward hierarchical clustering of the prokaryote community at the ASV level. Shapes correspond to the size fraction and colors to the sampling group (as identified in panel c). Sample identification corresponds to panel b. (b) Stacked bar graph of the read proportions of prokaryote ASVs at class level. Samples are ordered according to the hierarchical clustering in panel a. (c) Bar graphs showing the mean Chao1 index of prokaryotes for each sampling group and separated by size fraction (darker colors and circles correspond to large size fraction; lighter colors and triangles correspond to small size fraction). Shapes correspond to individual values, and error bars are SD. $n = 3$ except for *R* large size fraction ($n = 2$). Site labels are as in Table 1 and Fig. 2 legends. Sample *R.3* (>3 µm) is missing as the PCR amplification was unsuccessful.

sampling group. Overall, the plume microbial community clustered with the deeper river site, while brackish and shallow river samples clustered with their respective replicates. Separation by size fraction (permutational MANOVA, prokaryotes $R^2 = 0.24$, $P < 0.001$, microbial eukaryotes $R^2 = 0.12$, $P = 0.02$; homogeneity of variance in the dispersion matrix, prokaryotes $P = 0.12$, microbial eukaryotes $P = 0.74$) and by sampling groups (permutational MANOVA, prokaryotes $R^2 = 0.27$, $P < 0.001$, microbial eukaryotes $R^2 = 0.59$, $P < 0.001$; homogeneity of variance of the dispersion matrix, prokaryotes $P = 0.43$, microbial eukaryotes $P = 0.97$) was confirmed by a permutational multivariate analysis of variance.

The prokaryote community (Fig. 3b; Fig. S2) was dominated by Gammaproteobacteria (mostly *Polynucleobacter* and unclassified Gammaproteobacteria), Alphaproteobacteria (mostly SAR11 clade Ia and *Planktomarina* for the brackish samples and SAR11 clade III for all sampling groups), Bacteroidia (mostly *Sediminibacterium* and *Fluviicola*, for all sampling groups and unclassified NS9 marine group and Crocinitomicaceae for the brackish samples), Verrucomicrobiae (mostly *Chthoniobacter* and unclassified Verruco-microbiae), and Actinobacteria (mostly hgcI clade and unclassified Sporichthyaceae). Unclassified OM190 (mostly in brackish samples), as well as CL500-29 marine group, *Cyanobium PCC-6307,* and CL500-3 were also among the most abundant taxa. Archaeal ASVs represented less than 0.1% of prokaryote reads at most sites but accounted for 0.78%–8.47% in brackish samples, mainly attributed to the genus *Nitrosopumilus*.

The microbial eukaryote (Fig. 4b; Fig. S3) community was dominated by Dinoflagellata for the brackish samples, while Ciliophora and Ochrophyta dominated the other sites with a greater relative abundance of the former in the shallow river sites and of the latter in the plume and deeper river sites. The most abundant taxa of the microbial eukaryote community also included, among others, the division Telonemia, Perkinsea, and Katablepharidophyta.

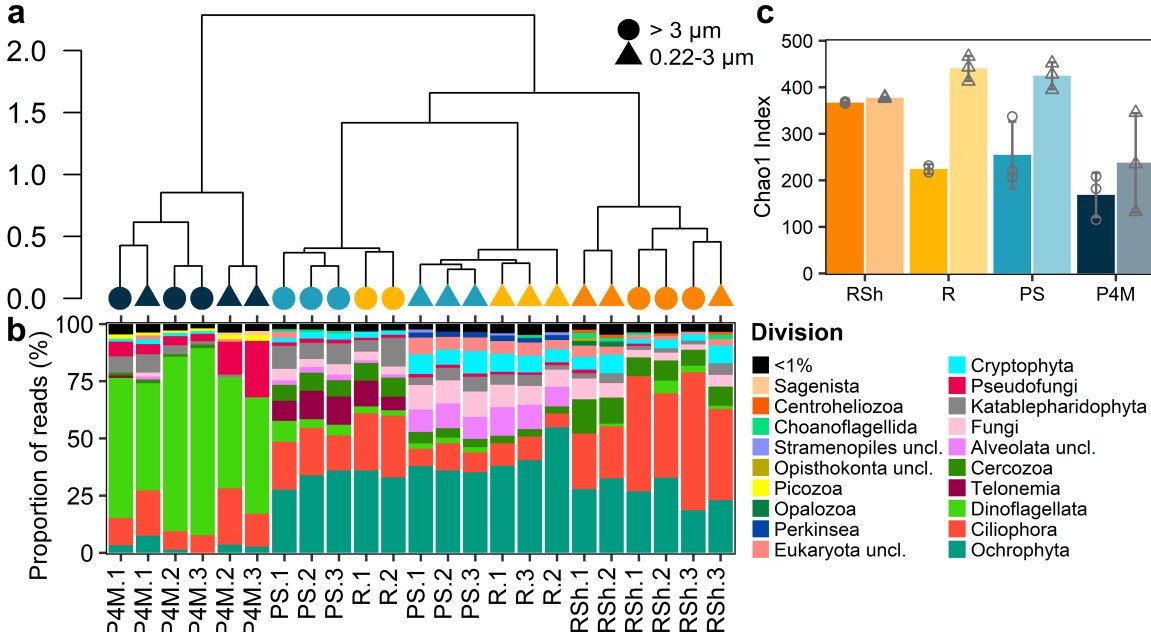

**FIG 4** Eukaryotic communities in the Great Whale River and Plume. (a) Ward hierarchical clustering of the microbial eukaryote community at the ASV level. Shapes correspond to the size fraction and colors to the sampling group (as identified in panel c). Sample identification corresponds to panel b. (b) Stacked bar graph of the read proportions of microbial eukaryote ASVs at division level. Samples are ordered according to the hierarchical clustering in panel a. (c) Bar graphs showing the mean Chao1 index of microbial eukaryotes for each sampling group and separated by size fraction (darker colors and circles correspond to large size fraction; lighter colors and triangles correspond to small size fraction). Shapes correspond to individual values, and error bars are SD. $n = 3$ except for *R* large size fraction ($n = 2$). Site labels are as in Table 1 and Fig. 2 legends. Sample *R.3* (>3 μm) is missing as the PCR amplification was unsuccessful.

Alpha-diversity (Fig. 3c and Fig. 4c) differed between size fractions for both the prokaryotes and microbial eukaryotes (Wilcoxon signed-rank test for paired samples, $P < 0.001$ in both cases) and was greater for the large size fraction for the prokaryotes and the small size fraction for the microbial eukaryotes.

## Metagenomic analysis

Metagenome coassembly resulted in 3,620,822 contigs with an N50 of 619 pb and an average read mapping of 74% (71.8% ± 1% for *RSh* sites, 80.7% ± 1% for *R*, 83.1% ± 1% for *PS*, and 63% ± 5% for *P4M*). A total of 4,965,927 open reading frames were identified with 1,828,305 coding DNA sequences annotated with the KEGG database resulting in 12,773 KOs. Functional assignments of the reads without assembly resulted in 15,561 KOs. Most of the KOs (73% for the coassembly and 72% for the reads) were present in all sampling groups. Hierarchical clustering of functional beta-diversity (Fig. S4) revealed the same general clustering as for the community composition, except one brackish sample that was more similar to the plume/deeper river group (permutational MANOVA by sampling groups, for the coassembly $R^2 = 0.67$, $P < 0.001$, for the direct reads annotation $R^2 = 0.75$, $P < 0.001$; homogeneity of variance of the dispersion matrix, coassembly $P = 0.23$, reads annotation $P = 0.16$).

## Overview of metabolic potential

The metagenomes were analyzed to evaluate the metabolic potential of the winter microbial community, with a focus on nitrogen, carbon, and sulfur metabolism, as well as photosynthesis (Fig. 5). Table S1 contains a list of all genes attributed to the pathways/reactions described below. Genes associated with these metabolic pathways that were significantly ($P_a \leq 0.01$) differentially abundant between the two river sampling groups (shallow vs deeper sites, Fig. 6a; Table S2) and/or along the vertical plume profile

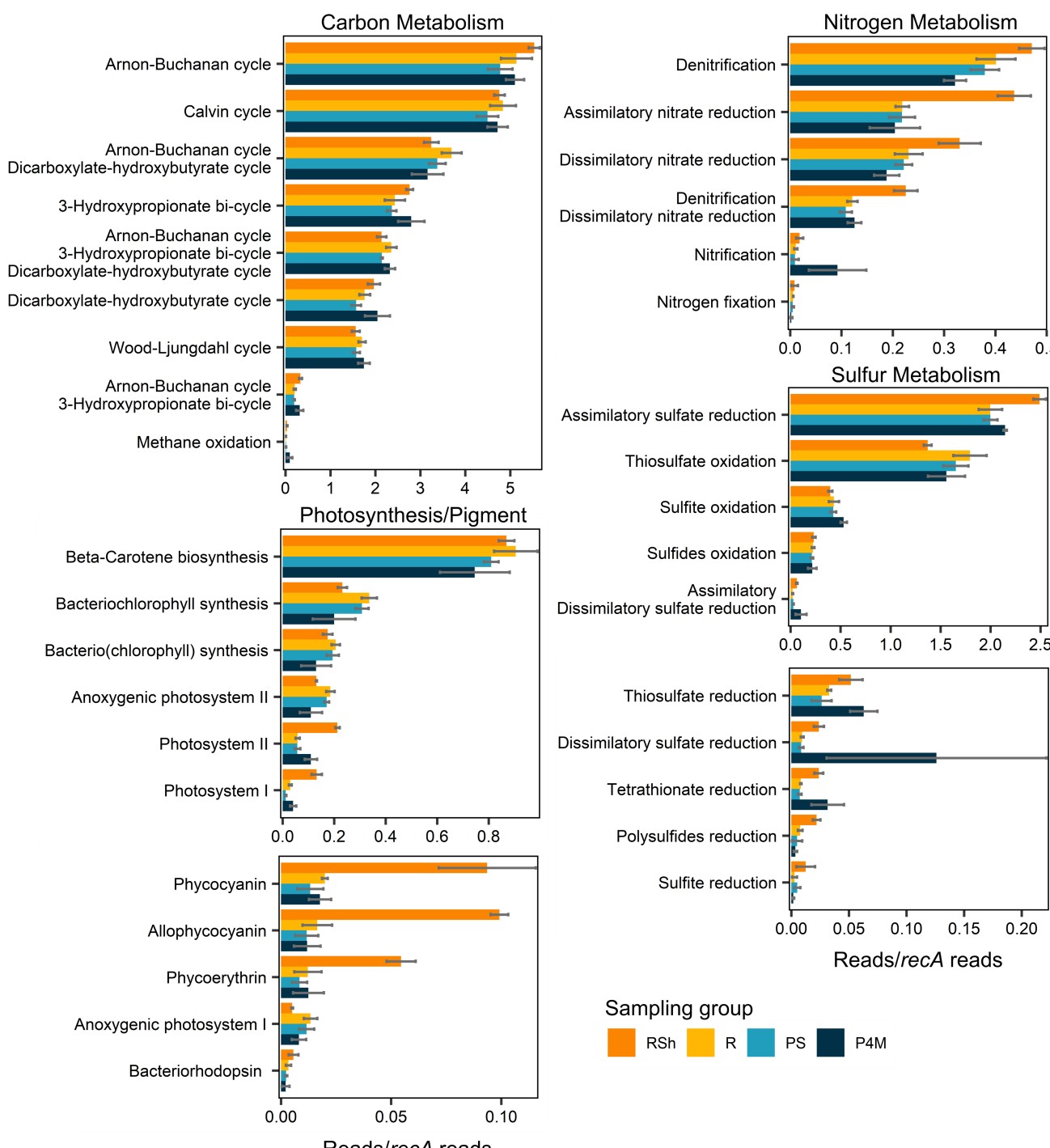

**FIG 5** Means ± SD of the sum of normalized gene abundances (reads/*recA* reads) associated with pathways/reactions for carbon, nitrogen, and sulfur metabolism and for photosynthesis for each sampling group. The list of genes used is provided in Table S1. Bars with multiple pathways/reaction names consist of KOs shared between these pathways/reactions, except *pmo-amoABC* genes, which were added to the sum of reads for nitrification and methane oxidation. Significantly differentially abundant genes (adjusted *P*-values ≤0.01) between shallow and deeper river sites and between plume surface and brackish water are presented in Fig. 6a and b.

(surface low salinity vs brackish, Fig. 6b; Table S3) were also identified. In this section, when genes are indicated to be differentially abundant in one of the sampling groups, we refer to these identified genes.

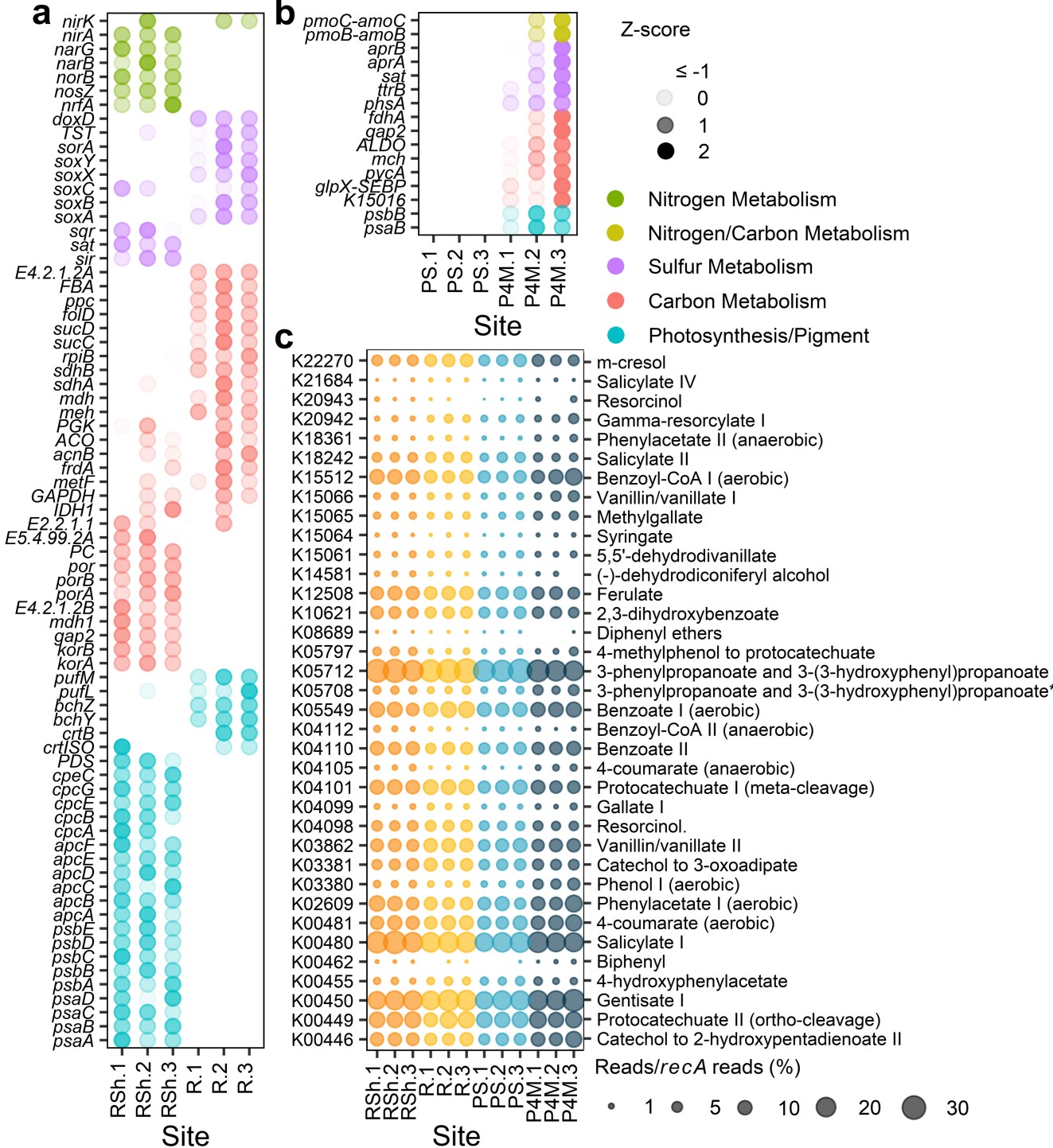

**FIG 6** (a and b) *Z*-score (SD from the row mean, calculated from normalized gene abundance reads/*recA* reads) of significantly differentially abundant genes (adjusted *P*-values ≤0.01) between shallow and deeper river sites (*RSh* vs *R*; panel a) and between plume surface and brackish water (*PS* vs *P4M*; panel b). The genes shown are restricted to those implicated in the reactions outlined in Fig. 5 for nitrogen, carbon, and sulfur metabolism and photosynthesis/pigment. (c) Bubble plot of the abundance (reads/*recA* reads; %) of marker genes implicated in the degradation of aromatic compounds. Marker genes are identified by their KEGG number and name of the degradation pathway. 3-Phenylpropanoate and 3-(3-hydroxyphenyl)propanoate* is short for 3-phenylpropanoate and 3-(3-hydroxyphenyl)propanoate to 2-hydroxypentadienoate. Color represents different sampling groups [*RSh,* shallow river sites; *R,* deeper river sites; *PS,* plume surface; *P4M,* plume at 4 m depth (brackish water)].

Many genes associated with the nitrogen cycle were identified. Nitrogen fixation genes (*nifD*, *nifH*, *nifK*, *nifB*, *nifE*, *nifN*, *nifX*, and *nifZ*) were present in some of the samples but at very low abundance. Nitrification genes (*hao*, *amoA*, *amoB*, *amoC*, *nxrA,* and *nxrB*) were detected in all sampling groups, with a greater abundance of *amoB* and *amoC* in the brackish sites. The presence of the genus *Nitrospira* in the amplicon data set and the taxonomic assignment to the phylum Nitrospirota of some of the contigs with genes encoding for hydroxylamine dehydrogenase and ammonia monooxygenase (Fig. S5a and b) suggested a potential for complete ammonia oxidation. Genes implicated in denitrification (including *nirK*, *nirS*, *nosZ*, *norC*, and *norB*) and in dissimilatory (*nirB*, *nirD*, *nrfA*, and *nrfH*) and assimilatory (*nasB*, *nasC*, *narB*, *NR*, and *nirA*) nitrate reduction to ammonium were widespread, and many were in greater abundance in the shallow river sites.

The KEGG database does not differentiate between genes encoding for methane and ammonia monooxygenase (*pmoA-amoA*, *pmoB-amoB*, and *pmoC-amoC*). However, some of the *pmo* genes were classified to the methanotrophic order Methylococcales (Fig. S5c), and the presence of other genes involved in methane oxidation (primarily *mdh1*, with *mmoX* and *mmoC* present in only some of the samples) indicated a potential for methanotrophy within the microbial community. Genes encoding for the methanogenesis key enzyme methyl-coenzyme M reductase were detected in very few samples and only through direct annotation of the reads and not in the coassembly.

Several autotrophic carbon fixation pathways were identified in the metagenomes, notably the Calvin-Benson cycle, the Arnon-Buchanan cycle, the Wood-Ljungdahl cycle, the 3-hydroxypropionate bi-cycle, and the dicarboxylate-hydroxybutyrate cycle. All KOs identified as essential in the KEGG mapper were present in the coassembly data set, except K14467 for the dicarboxylate-hydroxybutyrate cycle. This KO, however, was detected in a few samples through direct annotation of the reads. Despite the differential abundance of many genes associated with autotrophic carbon fixation pathways between our sampling groups, no pathway appeared to be preferentially favored in any of the groups.

Genes for sulfur metabolism were numerous, with the potential for oxidation and reduction of S-compounds including sulfides (oxidation: *fccA*, *fccB*, and *sqr*), polysulfides (reduction: *hydA*, *hydB*, *hydD*, and *hydG*), thiosulfate (oxidation: *soxA*, *soxB*, *soxC*, *soxX*, *soxY*, *soxZ*, *TST*, *doxA*, and *doxD*; reduction: *phsA* and *phsC*), tetrathionate (reduction: *ttrA*, *ttrB*, and *ttrC*), and sulfite (oxidation: *soeA*, *soeB*, *soeC*, *sorA*, and *SUOX*; reduction: *asrA* and *asrB*). Genes encoding for assimilatory (*cysC*, *cysD*, *cysH*, *cysI*, *cysJ*, *cysN*, *cysNC*, *PAPSS*, *sat*, and *sir*) and dissimilatory (*aprA*, *aprB*, *dsrA*, *dsrB,* and *sat*) sulfate reduction to sulfide were also present. In the river, most of the genes involved in thiosulfate oxidation were more abundant in the deeper sites. Deeper river sites also had greater abundance of a gene involved in sulfite oxidation (*sorA*), while the shallow sites had greater abundance of genes involved in assimilatory sulfate reduction (*sir*, *sat*) and in the oxidation of sulfides (*sqr*). The brackish water samples had a greater abundance of genes encoding for dissimilatory sulfate reduction (*aprA*, *aprB*, *sat*) and reduction of thiosulfate (*phsA*) and tetrathionate (*ttrB*).

There was evidence of photosynthetic potential with genes involved in photosystems I and II and genes encoding photosynthetic antenna proteins for the cyanobacterial pigments allophycocyanin (*ApcA* to *ApcF*), phycocyanin (*CpcA* to *CpcG*), and phycoerythrin (*CpeB* to *CpeE*, *CpeS*, *CpeT*, *CpeU*, *CpeY*, *CpeZ*). Genes for the photoprotective pigment beta-carotene were also present (*ZDS*, *crtB*, *PDS*, *lcyB*, *crtISO*, *Z-ISO*). However, most of the genes for the light-harvesting chlorophyll protein complexes I and II that are characteristic of green algae were absent from most samples. Key genes involved in the synthesis of (bacterio)chlorophyll (*chlP/bchP, chlG/bchG*), bacteriochlorophyll (*bchX, bchY, bchZ*), and bacteriorhodopsin (*bop*) were present in all samples, and so were the genes for anoxygenic photosystems I (*pscC*) and II (*pufl, pufM*). The shallow river sites appeared to have greater potential for photosynthesis, with a greater abundance of genes for photosystems I and II, allophycocyanin, phycocyanin, and phycoerythrin. In comparison,

deeper river sites had a greater abundance of genes involved in anoxygenic photosystem II and bacteriochlorophyll synthesis.

## Potential to degrade aromatic compounds

Numerous marker genes for aromatic degradation pathways that have been previously identified in the Canadian Basin of the Arctic Ocean (14) were present in the metagenomes (Fig. 6c). The most abundant were for the degradation of gentisate I (E1.13.11.4), salicylate I (E1.14.13.1), and 3-phenylproponate and 3-(3-hydroxyphenyl)propanoate (*mhpA*). Marker genes for the anaerobic degradation of gallate III (*lpdC*) and for the degradation of pinoresinol (*pinA*) and γ-resorcylate II (*tsdB*) were absent from our metagenomes. In the river, shallow sites had a greater abundance of the marker gene associated with the degradation of 4-methylphenol to protocatechuate (*pchF*), while the deeper river sites had greater abundances for degradation pathways of gentisate I, vanillin and vanillate II (*vanA*), γ-resorcylate I (*graA*), catechol to 3-oxoadipate (*catA*), and m-cresol (*nagX*). In the plume vertical profile, the brackish water community had greater potential for aerobic degradation of phenol I (E1.14.13.7) and anaerobic degradation of 4-coumarate (*hbaA*).

## DISCUSSION

The logistical challenges posed by winter sampling (56) and the previously held view of negligible biological activity during winter (57) have resulted in a scarcity of data and understanding about the structure and functioning of aquatic ecosystems during winter relative to summer. This is especially the case for seasonally ice-covered lotic environments (58), limiting our understanding of these ecosystems. In the present study, which extended a previous work characterizing the summer community composition of the Great Whale River and surrounding waters (20), we investigated the composition and metabolic capabilities of the winter under-ice microbiome of the Great Whale River and its discharge plume into Hudson Bay. We found substantial diversity in both community composition and functional capabilities, including the capacity to degrade complex terrestrial compounds, despite the constraints imposed by seasonal ice-cover and near-freezing water temperatures.

### Microbial community composition and diversity

Microbial community composition and richness differed by size fraction, as is often the case (59, 60), reflecting community differences in lifestyle for the prokaryotes and in the size range of microbial eukaryotes. Although particulate organic and inorganic matter concentrations in the Great Whale River have been reported to be lower in winter (61), the richness of the prokaryote community was greater for the large size fraction. This indicates that particles continued to offer heterogeneous microhabitats and niches for bacteria in the water column under the ice as they do in open water (59, 62, 63), despite the reduced runoff from the watershed that result in lower particle concentrations and likely decreased heterogeneity among particles in terms of size and composition.

For microbial eukaryotes, the greater richness of the small size fraction contrasted with the previously reported richness pattern of the summer community, for which there were no differences between size fractions (20). This seasonal difference would be consistent with a better acclimation of picoeukaryotes to low light and temperature, favoring them during the winter season (64). It may also indicate seasonal transitions for certain microbial eukaryotes that exhibit size variations throughout various stages of their life cycle (e.g., gametes, vegetative cells, and spores). Processes such as asexual division or transition to a quiescent stage can also result in a decrease of cell size (65, 66).

Community composition also varied among the sample groups in association with changes in the limnological variables. For the river, these changes were related to the proximity of the sediments (riverbed) to the ice-cover, while in the plume profile, they were related to the transition to a different water mass with higher salinity. At the

shallow depth sites, the influence of resuspended near-bottom sediments, including bed load particles (67), was more pronounced, likely due to the faster flowing waters in the reduced water column between the lower ice surface and the riverbed. In these sites, there was a greater relative abundance of ciliates, including the suspension-feeding genus *Vorticella*, which commonly takes on a sessile form that attaches to various substrates (68). This suggests an increased immigration of benthic microorganisms into the pelagic community of shallow waters. In addition, many of the most abundant taxa that were in greater proportion in shallow sites are found in soils, sediment, or groundwater including the genus *Candidatus* Solibacter [Acidobacteriae; (69)] and the bactivorous genus *Sandona* [Cercozoa; (70)].

Coastal Hudson Bay is strongly influenced by river runoff, which creates buoyant freshwater plumes that spread out over the marine waters. During the winter season, the depth and extent of these plumes increase due to the presence of an ice cover that impedes wind-induced mixing with the underlying seawater (15, 71). At the time of the sampling, the vertical transition from the Great Whale River freshwater to the marine Hudson Bay water was around 4 m depth as indicated by the rapid increase in salinity which went from 0.3 PSU at 3.75 m to 24.3 PSU at 4.5 m depth. In a summer sampling of the coastal region of Hudson Bay near the Great Whale River, it was reported that a coastal site with a salinity (25.33 PSU) similar to the salinity of the bottom layer in our vertical profile (24.5 PSU) had a microbial eukaryote community composed entirely of marine taxa (19). Here, at 4 m deep, the community consisted of freshwater and marine taxa, indicative of the mixing of the two water masses. The prokaryote community was dominated by freshwater taxa, while the microbial eukaryotes were dominated by marine taxa, mainly Dinoflagellata but also MAST-1A and the ciliate *Mesodinium*, all commonly detected in the summer community of coastal Hudson Bay (19). Among the marine prokaryotes present in the brackish samples was the SAR11 clade Ia (Alphaproteobacteria), a widely distributed and abundant clade in ocean surface waters that plays a major role in the carbon cycle (72, 73). Additionally, there were marine taxa that appear to thrive under winter conditions, as their relative abundances have been reported to increase in winter in some studies, such as the genus *Planktomarina* (Alphaproteobacteria), which is more abundant in the Ofunato Bay (Japan) during winter (74) and the ammonia-oxidizing archaeon *Nitrosopumilus* (75), which increases in abundance in winter in the west coast of the Antarctic Peninsula (76), in the Arctic Ocean (77), and in the coastal North Sea (78).

The abundance of prokaryotes (ranging from 3.6 to $9.7 \times 10^5$ cells mL$^{-1}$) and especially phytoplankton (ranging from 0.3 to $2.5 \times 10^3$ cells mL$^{-1}$, 0.06 to 0.86 µg chlorophyll-*a* L$^{-1}$) was greatly reduced compared to the previously reported summer values [prokaryotes 1.8 to $3.1 \times 10^6$ cells mL$^{-1}$, phytoplankton 9.3 to $21.4 \times 10^3$ cells mL$^{-1}$, chlorophyll-*a* 0.46 to 1.22 µg L$^{-1}$; (20)]. Photoautotrophic taxa were not dominant in the microbial eukaryote community. This was to be expected as snow covering the ice would greatly reduce the photosynthetically available radiation in the water column (79, 80). Punctual exposure to light was, however, possible for the microbial community due to the lack of ice over some sections of the river a few kilometers upstream of the shallow river sites and the absence of snow covering the clear ice in some areas of coastal Hudson Bay.

In lakes, winter blooms of cyanobacteria are reported in the absence of snow covering the ice [e.g., references (81, 82)], but the presence of an ice cover can suppress the growth of picocyanobacteria (83). In the Great Whale River, cyanobacteria, mainly the genus *Cyanobium PCC-6307*, accounted for a non-negligible part of the community with a relative abundance of up to 3%. This contrasts with the lower relative abundance in summer [up to 1% of the reads; (20)], although caution should be used when comparing relative abundance, as changes in one taxon may be due to changes in total population size of the overall community. The greater relative abundance of cyanobacteria in the shallow river sites determined by amplicon data was consistent with the metagenomic analysis showing a greater abundance of genes associated with cyanobacterial pigments such as allophycocyanin, phycocyanin, and phycoerythrin at the same sites. It is also

likely that the cyanobacteria were responsible for the higher concentrations of chlorophyll-*a* and the greater abundance of genes associated with photosystems I and II in these sites. In Lake Tiefer (Germany), cyanobacterial abundance was lower in winter, but *Cyanobium* dominated the community with some ASVs specific to this season suggesting a cold-adapted lineage (84).

In the river, the microbial eukaryote community was predominantly composed of taxonomic groups such as Ciliophora and Chrysophyceae (Ochrophyta) for which mixotrophy and heterotrophy are common features (85, 86). Phagotrophy by these taxa would provide a competitive advantage under the low-light conditions encountered in winter by providing alternative carbon and energy sources. Among these groups were the omnivorous ciliate *Urotricha* (87), the mixotrophic ciliate *Askenasia* (88), and *Spumella,* a colorless, heterotrophic chrysophyte (89). Additionally, predatory heterotrophs such as Telonemia (90) and Katablepharidophyta (91) were also relatively abundant in the community. Dominant taxonomic groups in the Great Whale River were similar to those reported in a seasonal study characterizing the microbial eukaryotic community of the Saint-Charles River (Quebec, Canada), in which higher proportions of Ciliophora, Chrysophyceae, and Telenomia were observed in winter [cold season; (8)]. In that study, the authors also reported a greater relative abundance of Cryptophyta and Dinoflagellata during the warm season (8). When comparing our winter data set to the previously published values for the summer community of the Great Whale River (20), we note a winter decline in the relative abundance of the Dinoflagellata (relative abundance ranging from 5% to 18% in summer, while up a maximum of 6% in winter) consistent with Cruaud et al. (8) for the Saint-Charles River. There was also a decrease for other taxa notably the parasite Perkinsea (up to 39% in summer but only up to 2.5% in winter), suggesting a seasonal variation of their host availability, and the heterotrophic Choanoflagellida (up to 19% in the summer compared to up to 2% in the winter), suggesting that not all heterotrophs are favored by winter conditions. Choanoflagellates were more dominant in winter in the sub-Arctic fjord Ramfjorden [Norway; (92)], but no seasonal difference was reported by Cruaud et al. (8), indicating that the decline we observed may be related to specific local conditions or different choanoflagellates community composition.

Many of the dominant bacterial genera in winter, including *Sediminibacterium* (Bacteroidia), the hgcl clade (also known as *acl* and Nanopelagicales; Actinobacteria), and *Polynucleobacter* (Gammaproteobacteria), were also dominant in the summer community (20). The latter two are also ubiquitous taxa in freshwater ecosystems that frequently constitute a significant proportion of the bacterial community (93–95). Their seasonal persistence and dominance likely reflect their ability to grow on both autochthonous and allochthonous carbon or to use alternative energy sources, as suggested in a Toolik Lake study (Alaska, USA), in which a large fraction of the bacterial community consisted of taxa that persisted throughout the year, including many cosmopolitan taxa (96).

## Microbial community metabolic potential

While most of the KOs were present in all samples, metabolic potential varied with the proximity of the sediment to the ice cover in the river and with the origin of the water mass (freshwater or marine) in the vertical plume profile, mirroring changes in community composition and environmental variables. Differences in the plume are consistent with previous comparative metagenomic analysis between freshwater and marine ecosystems that have found shared core functions between ecosystems but also differences in metabolic potential including in genes involved in osmoregulation, amino acid metabolism, and active transport (97).

The prevalence of genes associated with nitrogen fixation was lower than for nitrification and denitrification. Nitrogen fixation is energetically costly to perform (98), and a previous study of the Great Whale River found higher nitrate, ammonium, and TN concentrations during ice-covered periods (61), which would reduce the need for $N_2$ fixation. In lakes, ammonium concentrations are a strong predictor of winter nitrification

rates (99), suggesting that higher ammonium concentrations in the Great Whale River during the winter may create favorable conditions for nitrification. In coastal lagoons of the Alaskan Beaufort Sea region, nitrification genes were found to be in greater abundance in winter and were associated with an increasing relative abundance of Thaumarchaeota (10). Similarly in our study, genes encoding for ammonia monooxygenase were in greater abundance in the brackish sites, where the archaeaon *Nitrosopumilus* accounted for up to 6% of the prokaryote community. Genes implicated in denitrification and dissimilatory nitrate reduction were present in the well-oxygenated water column. The potential for these nitrogen cycle processes was greater in the shallow river sites, where there were higher TSS concentrations. Within the oxic water column, denitrification occurs in suspended particulate matter, where micro-habitat redox conditions can shift from oxic at the surface of a particle to anoxic in the center (100). Additionally, an experimental study found that the denitrification rates increase linearly with suspended particulate matter concentration (101), which could explain the abundance of these genes with increasing TSS concentration in our samples.

While anoxic micro-niches within particles may allow denitrification in the river and its plume, they did not seem to promote methanogenesis, as genes encoding for the methyl-coenzyme M reductase were missing from the water column samples (based on the coassembly results). As the basal member of the redox ladder, methanogenesis is less thermodynamically favorable (67), and microorganisms associated with more thermodynamically favorable reactions are expected to outcompete methanogens (102). Under the ice, methane concentrations of the Great Whale River and its plume are reported to be lower than the summer values but still higher than the atmospheric concentrations (103), suggesting advection from the sediments or inputs from tributaries. In the water column, there was the potential for oxidation of methane, notably by the genus *Methylobacter* that includes psychrophilic species (104), as genes involved in methanotrophy were present. Methanotrophy occurs in rivers in winter but at lower rates than during the ice-free period (105).

The winter community had the potential to use various inorganic sulfur compounds as energy sources and as both electron donors and acceptors. This aligns with previous findings on the presence (10, 13, 106) and expression (12) of genes involved in the sulfur cycle under ice-covered conditions. It is likely that anoxic sulfur processes, such as dissimilatory sulfate reduction, occur within suspended sediments, similar to denitrification. The higher potential for sulfur compound reduction (sulfate, thiosulfate, and tetrathionate) observed in the brackish samples may be attributed to the greater availability of sulfate, which is the most abundant electron acceptor in marine water. Genes involved in several autotrophic carbon fixation pathways were also present, consistent with the widespread genomic potential for carbon fixation among prokaryote taxa (107).

During winter freezing and ice cover, the supply of autochthonous organic matter from primary production and the input of allochthonous matter from the frozen watershed are expected to be low (58). The composition and lability of this organic matter also change during the winter. For example, in a Yukon River study, the aromaticity and molecular weight of the dissolved organic matter were shown to decrease during the ice-cover period with an increase in autochthonous organic matter, suggesting consumption of aromatic compounds and bacterial production (108). In the Great Whale River, the fraction of the DOC present in late winter has been characterized as mostly semi-labile by incubation experiments, with a biodegradable proportion similar to that in rivers flowing into the Arctic Ocean (109). Given this similarity, we aimed to investigate marker genes involved in the degradation of aromatic compounds that were previously identified for the Arctic Ocean (14), which is heavily influenced by terrigenous inputs from large rivers (110). Most of these genes (35 out of 39) were present in our metagenomes, indicative of the potential for the winter community to degrade various aromatic compounds. Among them, the most abundant (i.e., fraction of bacterial genomes in the metagenome with the marker genes, calculated by normalizing to the

*recA* gene) were also prevalent in the Arctic Ocean, although with a higher proportion in the Arctic Ocean. For the degradation of gentisate, the maximum value was 26% in our metagenome vs 65% in the Arctic Ocean; for salicylate, this was 27% vs 45%; and for 3-phenylproponate and 3-(3-hydroxyphenyl)propanoate, this was 31% vs 40%. These pronounced differences suggest that the degradation of aromatic compounds was less widespread among the prokaryotes in our winter community and that many may rely on other resource acquisition mechanisms or organic matter sources. This comparison rests on only a small set of genes and requires a more systematic study in the future. However, the magnitudes of these differences are striking and are contrary to our initial hypothesis. The results imply that the subarctic river microbiome relies on lower molecular weight and more labile organic carbon than the offshore ocean, where the more labile compounds may be largely removed before the riverine DOC arrives offshore. It is estimated that 30% of terrestrial DOC is removed along the shelves before entering the Arctic Ocean (111), and in the western Arctic Ocean, terrestrial DOC is estimated to be mineralized with a half-life of $7 \pm 3$ years, suggesting a more refractory composition (112).

## Conclusions

Our study provides insights into the winter microbiome of ice-covered subarctic rivers and associated coastal marine waters. It is likely that some microorganisms are in a quiescent state during the winter season and that some metabolic capabilities discussed here represent the functional potential of the community rather than functional activities at the time of sampling. This will require future assessment by metatranscriptomic analysis and by rate measurements of biogeochemical processes. However, the findings suggest that this cold-dwelling microbial community is capable of diverse metabolic pathways for energy and carbon acquisition, including phototrophy and the degradation of aromatic compounds. Additionally, the microbiome exhibited changes based on water masses and sediment proximity to the ice cover. The dominant taxa had been previously detected in summer, indicating their ability to persist over time in a range of physico-chemical conditions. There was evidence of large differences in the potential degradation of aromatic organic compound in the river vs offshore Arctic Ocean, indicating the need for much closer attention to terrigenous carbon cycling by high-latitude freshwater and ocean microbiomes.

## ACKNOWLEDGMENTS

We thank the communities of Kuujjuarapik and Whapmagoostui, Sydney Arruda for the help at the field station, our local guide Frederick Audlarock, Marie-Josée Martineau for technical assistance with the HPLC and for the chromatogram analysis, and Marianne Potvin for technical guidance regarding molecular work. We also thank Lise Rancourt and the Institut National de la Recherche Scientifique, Centre Eau-Terre-Environnement (INRS-ETE) for chemical analyses and the Plateforme d'Analyses Génomiques (IBIS, Laval University, Québec) for the amplicon and metagenome sequencing.

This research was supported by the Sentinel North program of Université Laval, funded by the Canada First Research Excellence Fund (CFREF). Additional funding and support were provided by the Natural Sciences and Engineering Research Council of Canada (NSERC), the Canada Research Chair program, the Canada Network of Excellence ArcticNet, and the Centre for Northern Studies (CEN). This is a contribution to the project Terrestrial Multidisciplinary distributed Observatories for the Study of Arctic Connections (T-MOSAiC), under the auspices of the International Arctic Science Committee (IASC).

## AUTHOR AFFILIATIONS

[1]Département de Biologie, Université Laval, Quebec City, Quebec, Canada
[2]Institut de Biologie Intégrative et des Systèmes (IBIS), Université Laval, Quebec City, Quebec, Canada

[3]Centre for Northern Studies (CEN), Université Laval, Quebec City, Quebec, Canada

[4]Takuvik Joint International Laboratory, Université Laval, Quebec City, Quebec, Canada

[5]Québec-Océan, Université Laval, Quebec City, Quebec, Canada

[6]Centro de Química Estrutural, Departamento de Engenharia Química, Instituto Superior Técnico, Universidade de Lisboa, Lisboa, Portugal

[7]Institute for Bioengineering and Biosciences, Instituto Superior Técnico, Universidade de Lisboa, Lisboa, Portugal

[8]Associate Laboratory i4HB—Institute for Health and Bioeconomy at Instituto Superior Técnico, Universidade de Lisboa, Lisboa, Portugal

## PRESENT ADDRESS

Marie-Amélie Blais, Département de Biologie, Université de Sherbrooke, Sherbrooke, Quebec, Canada

Alex Matveev, Department of Geography and Environment, Concordia University, Montréal, Quebec, Canada

Lígia Fonseca Coelho, Department of Astronomy, Cornell University, Ithaca, New York, USA

Lígia Fonseca Coelho, Carl Sagan Institute, Ithaca, New York, USA

## AUTHOR ORCIDs

Marie-Amélie Blais  http://orcid.org/0000-0002-7649-1964
Warwick F. Vincent  http://orcid.org/0000-0001-9055-1938
Adrien Vigneron  http://orcid.org/0000-0003-3552-8369
Aurélie Labarre  http://orcid.org/0000-0002-6709-0042
Alex Matveev  http://orcid.org/0000-0003-4103-9131
Lígia Fonseca Coelho  http://orcid.org/0000-0001-5008-1249
Connie Lovejoy  http://orcid.org/0000-0001-8027-2281

## FUNDING

| Funder | Grant(s) | Author(s) |
| --- | --- | --- |
| Canada First Research Excellence Fund (CFREF) | | Warwick F. Vincent |
| Canadian Government \| Natural Sciences and Engineering Research Council of Canada (NSERC) | | Warwick F. Vincent |
| ArcticNet | | Warwick F. Vincent |
| Canada Research Chairs (Chaires de recherche du Canada) | | Warwick F. Vincent |
| UL \| Sentinelle Nord, Université Laval (Sentinel North) | | Warwick F. Vincent |

## AUTHOR CONTRIBUTIONS

Marie-Amélie Blais, Conceptualization, Formal analysis, Investigation, Methodology, Software, Visualization, Writing – original draft, Writing – review and editing | Warwick F. Vincent, Conceptualization, Funding acquisition, Project administration, Resources, Supervision, Writing – review and editing | Adrien Vigneron, Methodology, Writing – review and editing | Aurélie Labarre, Software, Writing – review and editing | Alex Matveev, Investigation, Writing – review and editing | Lígia Fonseca Coelho, Investigation, Writing – review and editing | Connie Lovejoy, Resources, Supervision, Writing – review and editing

## DATA AVAILABILITY

The molecular data sets generated in this study are available in the NCBI online repository (https://www.ncbi.nlm.nih.gov/; under the bioproject PRJNA999265 for the amplicon and PRJNA999354 for the metagenomes).The

environmental data are deposited in the northern environmental data repository Nordicana D (http://www.cen.ulaval.ca/nordicanad/en_index.aspx; doi: 10.5885/45870CE-F5AD28ECCD834122 and doi: 10.5885/45660CE-8B92339884C146D0).

## ADDITIONAL FILES

The following material is available online.

### Supplemental Material

**Supplemental material (Spectrum04160-23-s0001.docx).** Supplemental figures and tables.

### Open Peer Review

**PEER REVIEW HISTORY (review-history.pdf).** An accounting of the reviewer comments and feedback.

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
