## [Reviewer comments · Microbiology Spectrum]

Microbiology Spectrum

Diverse winter communities and biogeochemical cycling potential in the under-ice microbial plankton of a subarctic river-to-sea continuum

Marie-Amelie Blais, Warwick Vincent, Adrien Vigneron, Aurélie Labarre, Alex Matveev, Lúgia Coelho, and Connie Lovejoy

Corresponding Author(s): Marie-Amelie Blais, Universite Laval

Review Timeline:

Submission Date:	December 8, 2023
Editorial Decision:	February 7, 2024
Revision Received:	February 28, 2024
Accepted:	March 5, 2024

Editor: Jianjun Wang

Reviewer(s): Disclosure of reviewer identity is with reference to reviewer comments included in decision letter(s). The following individuals involved in review of your submission have agreed to reveal their identity: Donglei Sun (Reviewer #1)

Transaction Report:

DOI: <https://doi.org/10.1128/spectrum.04160-23>

Re: Spectrum04160-23 (Diverse winter communities and biogeochemical cycling potential in the under-ice microbial plankton of a subarctic river-to-sea continuum)

Dear Dr. Marie-Amelie Blais:

Thank you for the privilege of reviewing your work. Below you will find my comments, instructions from the Spectrum editorial office, and the reviewer comments.

Revision Guidelines

Sincerely,
Jianjun Wang
Editor
Microbiology Spectrum

Reviewer #1 (Comments for the Author):

The authors analyzed microbial communities and metagenomic composition of winter samples from the Great Whale River and its plume into Hudson bay. The study provides insights into the winter microbiome of ice-covered subarctic rivers and associated coastal marine waters. Overall, the study is well-designed and the manuscript is written nicely. I only have several minor questions.

1. I can't find the methods for measuring the oxygen and salinity of the sample.

2. what happened to the >3um fraction of the "R.3" sample, as it is missing in Figure 3 and 4
3. The authors found prokaryote community composition differed primarily by size fraction and then by sampling group, but opposite for eukaryotes. They need to discuss this findings in more detail.
4. Important comparisons should be marked if they are significant.

Reviewer #2 (Comments for the Author):

The authors investigated diversity and community structure of bacteria and eukaryotes using metabarcoding approaches in two size fractions of plankton along an ice-covered estuarine gradient of a subarctic river. Meanwhile, they also examined geochemical cycling-relevant genes to infer functional potential using metagenomics. They found the bacterial community structure was different between <3 and > 3 micron size fractions, but not for eukaryotes of these two sizes. They concluded that the microbial communities of subarctic rivers and their associated discharge plumes retain a broad taxonomic and functional diversity throughout the year. This work contributes by documenting the microbial diversity and spatial changes in a very special environment and season. Although the manuscript is well written, there are flaws that well-known nutrients like dissolved inorganic nitrogen species, dissolved silicate, and phosphate data are lacking. Apart from that, several aspects can be improved during revisions.

There are discussions on environmental factors influencing community composition of bacteria and microeukaryotes (lines 448-479). However, this is largely descriptive, without any statistic support. Multivariate analysis such as RDA or CCA can be performed. In fact, correlations between environmental variables and alpha diversity should be performed to specify the points in discussion as well.

Lines 246, 352, 372, 375, 376 and so on: Statistical significance should be supplied for these comparisons.

Figure 5: Error bars have been annotated, but significance was lacking for all these comparisons.

"The prevalence and abundance of genes associated with the nitrogen cycle indicate that nitrogen fixation was likely absent during the winter, whereas nitrification and denitrification were likely operating. " This does not make sense to me. The abundance of nif genes in the metagenomes was relative, not absolute quantity. Also bearing in mind what you are looking into is the genetic potential, instead of the expressed genes (activities).

Responses to reviews of the manuscript: “Diverse winter communities and biogeochemical cycling potential in the under-ice microbial plankton of a subarctic river-to-sea continuum” by M.A. Blais, W.F. Vincent, A. Vigneron, A. Labarre, A. Matveev, L.F. Coelho, and C. Lovejoy, submitted to **Microbiology Spectrum**.

Reviewer 1:

The authors analyzed microbial communities and metagenomic composition of winter samples from the Great Whale River and its plume into Hudson bay. The study provides insights into the winter microbiome of ice-covered subarctic rivers and associated coastal marine waters. Overall, the study is well-designed and the manuscript is written nicely. I only have several minor questions.

Response: We thank the reviewer for their comments.

1. I can't find the methods for measuring the oxygen and salinity of the sample.

Response: The oxygen and salinity were measured in situ at the sampling sites, not in the samples, using an RBR Concerto logger (measuring conductivity, temperature, salinity, and depth) and a YSI EXO2 (temperature, salinity, and dissolved oxygen). We have added that the YSI measured temperature, salinity and dissolved oxygen (Lines 130-131).

2. what happened to the >3µm fraction of the "R.3" sample, as it is missing in Figure 3 and 4

Response: This fraction was missing from the figures because the PCR amplification of the 3-µm filter sample of site *R.3* was unsuccessful for both the prokaryote and microbial eukaryote primers (Lines 186-188). The following sentence was added to Figures 3 and 4 legend: “Sample *R.3* (> 3 µm) is missing as the PCR amplification was unsuccessful.”

3. The authors found prokaryote community composition differed primarily by size fraction and then by sampling group, but opposite for eukaryotes. They need to discuss this findings in more detail.

Response: We apologise for the confusion and have changed this sentence (Lines 267-268) to: “Hierarchical clustering revealed that prokaryote (Figure 3a) and microbial eukaryote (Figure 4a) community composition at the ASV level differed by size fraction and sampling group.” We have rephrased it because although the clustering indeed suggests that prokaryotes differed primarily by size fraction, the PERMANOVA results indicated a slightly higher separation by sampling group ($R^2=0.24$ for size fraction and $R^2=0.27$ for sampling group).

4. *Important comparisons should be marked if they are significant.*

Response: The limnological variables comparisons are descriptive given the low number of replicates per sampling group (three), which limits the statistical power of analyses such as ANOVA, as does the presence of missing values (see Lines 131-132 and 163-164).

In response to this review comment we have rephrased part of the text to clarify that when we indicate greater abundance of a gene in a sampling group, we are referring to the results of the differential abundance analysis of KOs that were involved in pathways, reactions and modules presented in Figure 5 and that were found to be significantly differentially abundant (adjusted p -values ≤ 0.01 ; tested with package DESeq2 that can handle low numbers of replicates) along the river between the shallow and the deeper sites (*RSh* vs. *R*), and in the vertical plume profile between surface water and brackish water at 4 m depth (*PS* vs. *P4M*).

- a) We added the threshold p -value (≤ 0.01) used to determine KOs for which the z-score was calculated, at Line 247.
- b) We changed the legend of Figure 6a and 6b, it now reads: “**Figure 1. a-b**) Z-score (SD from the row mean, calculated from normalized gene abundance reads/*recA* reads) of significantly differentially abundant genes (adjusted p -values ≤ 0.01) between shallow and deeper river sites (*RSh* vs *R*; panel **a**) and between plume surface and brackish water (*PS* vs *P4M*; panel **b**). The genes shown are restricted to those implicated in the reactions outlined in Figure 5 for nitrogen, carbon and sulfur metabolism and photosynthesis/pigment.”
- c) We added two supplementary tables (Supplementary Table S2 and S3) presenting the DESeq2 results for these KOs (log2fold change and adjusted p -values).
- d) To make the text easier to read, we opted not to include the p -value after every comparison as they were all significant. However, we have added the following sentence for clarification in the first paragraph of this result section: “In the following section, when genes are indicated to be differentially abundant in one of the sampling groups, we refer to these identified genes”. (Lines 312-313).

Reviewer 2:

The authors investigated diversity and community structure of bacteria and eukaryotes using metabarcoding approaches in two size fractions of plankton along an ice-covered estuarine gradient of a subarctic river. Meanwhile, they also examined geochemical cycling-relevant genes to infer functional potential using metagenomics. They found the bacterial community structure was different between <3 and > 3 micron size fractions, but not for eukaryotes of these two sizes.

Response: We thank the reviewer for this very good summary, but would like to clarify one point. We also found differences in microbial eukaryote communities, both in terms of community structure, as indicated by a significant PERMANOVA result (Lines 270-271 in the revised manuscript), and in terms of richness, as indicated by the Wilcoxon signed-rank test for paired samples result (Lines 291-292).

They concluded that the microbial communities of subarctic rivers and their associated discharge plumes retain a broad taxonomic and functional diversity throughout the year. This work contributes by documenting the microbial diversity and spatial changes in a very special environment and season. Although the manuscript is well written, there are flaws that well-known nutrients like dissolved inorganic nitrogen species, dissolved silicate, and phosphate data are lacking. Apart from that, several aspects can be improved during revisions.

Response: We thank the reviewer for all of their helpful comments and suggestions. Concerning the point about nutrients, we sampled for and measured total N and total P concentration, but in addition, as there were historical seasonal data available, we incorporated them to further support our discussion; for example, at Lines 494-495.

There are discussions on environmental factors influencing community composition of bacteria and microeukaryotes (lines 448-479). However, this is largely descriptive, without any statistic support. Multivariate analysis such as RDA or CCA can be performed. In fact, correlations between environmental variables and alpha diversity should be performed to specify the points in discussion as well.

Response: We agree that multivariate analysis would have added further support to our conclusions, but we were constrained by the availability of environmental data and cross-correlations between variables. Salinity, temperature, and dissolved oxygen concentrations were missing for *R.2* and *R.3* due to logistical constraints (Lines 131-132), as were dissolved organic carbon data for *RSh.1* and colored dissolved organic matter concentrations for *RSh.1*, *RSh.3*, *R.2*, *P4M.1*, as some bottles were damaged during their transportation to the main lab (Lines 163-164). In addition, there was a strong correlation among our limnological variables (notably between total phosphorus, total nitrogen, total suspended sediments and chlorophyll-*a*), which limited the number of variables that we could have used in a constrained ordination analysis (see Supplementary Figure S1 that has been added). We also clarify this point in the text by adding the following sentence in Lines 248-250: “Constrained ordinations (e.g., redundancy analysis) were not performed due to the missing limnological variables for some samples and the high correlations among the remaining variables suggesting that they are confounding (Supplementary Figure S1).” The missing limnological variables are also the reason why we limited the comparison of richness to the size fraction and refrained from testing differences between sampling groups.

Lines 246, 352, 372, 375, 376 and so on: Statistical significance should be supplied for these comparisons.

Response: We thank the reviewer for bringing this to our attention. We have therefore changed relevant sections of the text to improve understanding, specifically:

- a) Line 246, the z-score were calculated only for KOs that were involved in pathways, reactions and modules present in Figure 5 and that were found to be significantly differentially abundant (adjusted *p*-values ≤ 0.01 ; tested with package DESeq2) along the river between the shallow and the deeper sites (*RSh* vs. *R*), and in the vertical plume profile between surface water and brackish water at 4 m depth (*PS* vs. *P4M*). The threshold *p*-value (≤ 0.01) used was added (Line 247).
- b) We changed the legend of Figure 6a and 6b, which now reads: “**Figure 2. a-b** Z-score (SD from the row mean, calculated from normalized gene abundance reads/*recA* reads) of significantly differentially abundant genes between shallow and deeper river sites (adjusted *p*-values ≤ 0.01 ;

RSh vs R; panel **a**) and between plume surface and brackish water (*PS vs P4M*; panel **b**). The genes shown are restricted to those implicated in the reactions outlined in Figure 5 for nitrogen, carbon and sulfur metabolism and photosynthesis/pigments.”

- c) We added two supplementary tables (Supplementary Table S2 and S3) presenting the DESeq2 results for these KOs (log2fold change and adjusted *p*-values).
- d) To make the text easier to read, we opted not to include the *p*-value after every comparison (Lines 352 and so on) as they were all significant. However, we have added the following sentence for clarification in the first paragraph of this result section: “In the following section, when genes are indicated to be differentially abundant in one of the sampling groups, we refer to these identified genes”. (Lines 312-313).

Figure 5: Error bars have been annotated, but significance was lacking for all these comparisons.

Response: Figure 5 provides an overview of the pathways/reactions discussed in the results. Standard deviations were added to illustrate the variability of the sums within each sampling group. Due to the small sample size (3) per group, which resulted in low statistical power, we refrained from conducting statistical analysis to compare the sum of normalized gene abundances (reads/*recA* reads) between sampling groups. This is for this same reason we did not perform ANOVA on our limnological variables. Instead, we chose to assess the differential abundance of each KO involved in the pathways/reactions presented in Figure 5, as this analysis is suitable for low numbers of replicates. The following sentence was added to the legend of Figure 5: “Significantly differentially abundant genes (adjusted *p*-values ≤ 0.01) between shallow and deeper river sites and between plume surface and brackish water are presented in Figures 6a and 6b.”

"The prevalence and abundance of genes associated with the nitrogen cycle indicate that nitrogen fixation was likely absent during the winter, whereas nitrification and denitrification were likely operating. " This does not make sense to me. The abundance of nif genes in the metagenomes was relative, not absolute quantity. Also bearing in mind what you are looking into is the genetic potential, instead of the expressed genes (activities).

Response: We agree that this was overextending. The sentence was changed to “The prevalence of genes associated with nitrogen fixation was lower than for nitrification and denitrification.” (Lines 493-494).

Re: Spectrum04160-23R1 (Diverse winter communities and biogeochemical cycling potential in the under-ice microbial plankton of a subarctic river-to-sea continuum)

Dear Dr. Marie-Amelie Blais:

Thank you for your efforts in revision by following the reviewers' comments. Your manuscript has been accepted, and I am forwarding it to the ASM production staff for publication. Your paper will first be checked to make sure all elements meet the technical requirements. ASM staff will contact you if anything needs to be revised before copyediting and production can begin. Otherwise, you will be notified when your proofs are ready to be viewed.

Sincerely,
Jianjun Wang
Editor
Microbiology Spectrum